# Reconstruction of Segmental Bone Defect in Canine Tibia Model Utilizing Bi-Phasic Scaffold: Pilot Study

**DOI:** 10.3390/ijms25094604

**Published:** 2024-04-23

**Authors:** Dae-Won Haam, Chun-Sik Bae, Jong-Min Kim, Sung-Yun Hann, Chang-Min Richard Yim, Hong-Seok Moon, Daniel S. Oh

**Affiliations:** 1Department of Prosthodontics, College of Dentistry, Yonsei University, Seoul 03722, Republic of Korea; drhaamdds@gmail.com; 2College of Veterinary Medicine, Chonnam National University, Gwangju 61186, Republic of Korea; csbae210@jnu.ac.kr; 3College of Health and Medical Sciences, Cheongju University, Cheongju 28503, Republic of Korea; vinaka00@gmail.com; 4Department of Precision Mechanical Engineering, Kyungpook National University, Sangju 37224, Republic of Korea; syhann@knu.ac.kr; 5School of Dental Medicine, Rutgers University, Newark, NJ 07102, USA; richard.yim@rutgers.edu; 6Department of Dental Biomaterials, College of Dentistry, Yonsei University, Seoul 03722, Republic of Korea

**Keywords:** segmental bone defect, canine model, hydroxyapatite, ceramic, scaffold, reconstruction

## Abstract

The reunion and restoration of large segmental bone defects pose significant clinical challenges. Conventional strategies primarily involve the combination of bone scaffolds with seeded cells and/or growth factors to regulate osteogenesis and angiogenesis. However, these therapies face inherent issues related to immunogenicity, tumorigenesis, bioactivity, and off-the-shelf transplantation. The biogenic micro-environment created by implanted bone grafts plays a crucial role in initiating the bone regeneration cascade. To address this, a highly porous bi-phasic ceramic synthetic bone graft, composed of hydroxyapatite (HA) and alumina (Al), was developed. This graft was employed to repair critical segmental defects, involving the creation of a 2 cm segmental defect in a canine tibia. The assessment of bone regeneration within the synthetic bone graft post-healing was conducted using scintigraphy, micro-CT, histology, and dynamic histomorphometry. The technique yielded pore sizes in the range of 230–430 μm as primary pores, 40–70 μm as secondary inner microchannels, and 200–400 nm as tertiary submicron surface holes. These three components are designed to mimic trabecular bone networks and to provide body fluid adsorption, diffusion, a nutritional supply, communication around the cells, and cell anchorage. The overall porosity was measured at 82.61 ± 1.28%. Both micro-CT imaging and histological analysis provided substantial evidence of robust bone formation and the successful reunion of the critical defect. Furthermore, an histology revealed the presence of vascularization within the newly formed bone area, clearly demonstrating trabecular and cortical bone formation at the 8-week mark post-implantation.

## 1. Introduction

Restoring segmental bony defects can present significant challenges due to various medical obstacles, because of the potential complications and poor prognosis associated with this type of injury. The impact on the patient is often significant and may result in permanent disability or loss of limb. Treatment recommendations for limbs, including long-bone defects, have included autologous bone grafting, bone transport, acute bone shortening, and amputation [1,2,3,4,5]. Autogenous bone grafting from the iliac crest or proximal tibia has become the gold standard in the treatment of “small” (<5 cm in humans) bony defects. However, donor site morbidity remains a significant concern with this technique. Larger, so-called “critical size” defects (>8 cm in humans, or >3 cm in canine models) will often fail to heal, despite multiple non-vascularized graft attempts [6,7,8,9].

Besides autogenous bone grafts, allografts relying on donor tissue, xenografts sourced from different species, and synthetic bone grafts composed of materials such as ceramics or polymers have been explored to address substantial bone defects. Additionally, strategies involving the use of bone morphogenetic proteins (BMPs) to stimulate growth, tissue engineering to create functional bone tissue in laboratories, stem cell therapy employing cells like mesenchymal stem cells (MSCs), platelet-rich plasma (PRP) containing growth factors, and nanotechnology for precise delivery and stimulation have all undergone investigation. Among these techniques, synthetic bone grafts have shown progress in addressing segmental bone defects, yet they are not without limitations. These limitations encompass challenges in achieving seamless integration and compatibility with natural bone, disparities in mechanical properties that may lead to potential stress-related issues, inadequate osteo-inductive potential for effective bone regeneration, concerns regarding regulatory approval and long-term efficacy, a lack of remodeling ability, potential immunological responses, and limited clinical evidence for complex scenarios. Ongoing research endeavors seek to mitigate these limitations; however, it remains imperative to carefully weigh the advantages and disadvantages when considering the use of synthetic grafts for addressing segmental bone defects [10,11,12].

The objective of this study is to test the feasibility of a synthetic bone graft composed of alumina (Al) and hydroxyapatite (HA), without adding cells or growth factors, to restore a segmental bony defect. Due to its biocompatibility, Al has been widely employed in various applications of orthopedic prostheses. Alumina’s exceptional chemical inertness, wear resistance, capability for attaining a highly polished surface finish, and notable hardness render it an apt selection for biomaterial use in load-bearing regions [13,14,15]. Also, HA has been used for a long period of time in the medical field as a bone graft or coating agent on metal implants. Consequently, the incorporation of both HA and Al into bi-phasic scaffolds within this study represents a promising avenue for evaluating the restoration of segmental bone defects.

## 2. Results

### 2.1. Scaffold Characterization

Figure 1 provides a comprehensive overview of the HA/Al bi-phasic scaffold. The fabricated scaffolds possessed specific dimensions, with an outer diameter measuring 1 cm, an inner diameter cavity of 3 mm, and a length of 2 cm, maintaining an anatomically confirmed structure (Figure 1). Micro-CT images in Figure 1B,D illustrate the presence of fully interconnected pores and a central cavity traversing the entire scaffold. The architectural composition of open pores and interconnected trabecular bone-like structures is evident in the SEM image presented in Figure 2A. The primary pores exhibited varying pore sizes ranging from 230 to 430 μm. Figure 2B highlights a secondary micro-channel structure, each with a diameter ranging from 40 to 70 μm, situated within every strut. Furthermore, Figure 2C depicts tertiary submicron holes, ranging from 200 to 400 nm in diameter, on the surface of the struts. Notably, all three of these structures exhibit interconnectivity, contributing to a collective porosity measuring 82.61 ± 1.28%, with a compressive strength equal to about 6.5 MPa. By conducting a crystalline phase analysis using XRD and EDS, as depicted in Figure 3, the inner alumina phase and outer HA phase was determined. In this study, the Ca/P ratio of stoichiometric HA was determined to be 1.64. These findings affirm the successful realization of a porous HA/Al bi-phasic scaffold through the utilization of a polymer template-coating technique.

### 2.2. Surgical Procedure and Radiographic Observation

Figure 4A,B provide a visual representation of a surgically induced 2 cm segmented defect in a canine tibia, accompanied by the implantation of the scaffold within the defect (Figure 4, yellow arrowhead). During the study, all subjects displayed sustained vitality, devoid of any signs indicating inflammation or infection at both the control and implantation sites. Remarkably, bone regeneration was observed, effectively bridging the defect and culminating in the formation of a new cortex within the illustrated area (Figure 5). A notably satisfactory integration between the scaffolds and the host bone was evident at both ends of the defect. At the 8-week juncture following implantation, a complete reunion of the defect was observed, encompassing the scaffold. Radiographic imaging was conducted at intervals of 0, 2, 4, 6, and 8 weeks post-surgery to assess the progression of bone healing and reunion within the defects (Figure 5). The bone healing was evaluated based on the radiographic bone healing index (Table 1) and was rated as 5 at 8 weeks after surgery.

In the control group, where the defects were left unfilled (Figure 4C, blue arrowhead), minimal bone growth from the host tissue were evident at 8 weeks after surgery (Figure 5(A0–A8)). The extremities of the bone defects exhibited sclerosis, and the medullary cavities remained obstructed at the 8-week post-surgery interval, indicating an absence of repair. In contrast, the defects that were filled with scaffolds showcased the onset of new bone formation after 2 weeks (Figure 5(B2)), with the scaffold still identifiable within the mid-region of the defect. By the 8-week milestone, new bone formation had occurred at the interfacial region between the bone graft and the host tissue, signifying a merger at both the proximal and distal host bone to the implant interfaces (Figure 5(B8)). The emergence of callus formation was notable at the periphery of the scaffold and along the contiguous host bone at 8 weeks post-surgery, indicating the development of mineralized tissue within the scaffold’s pores, including within the micro-channels. A 3D reconstructed CT image offered a clear depiction of the near-complete restoration of the original bone contour within the scaffold, affirming the comprehensive healing of the scaffold-filled defect site by 8 weeks post-surgery (Figure 5(B8-1)).

### 2.3. Scintigraphic Evaluation

The segmental defect evaluation with scaffolds included bony scintigraphy at 0, 4, and 8 weeks post-surgery. The utilization of radioactive tracers in bone scintigraphy has gained widespread acceptance for visualizing blood flow and bone metabolism. Among these tracers, 99mTc-HDP, a highly sensitive marker, is frequently employed in clinical bone research to gauge blood flow and metabolic activity in bone tissue [16,17]. This technique is deemed a gold standard for representing successful grafting, reliant on efficient delivery and an active osteocyte network [18]. In our study, we examined delayed images to unveil the bone metabolism and implant vascularization. Following the injection of 99mTc-HDP, half of the tracer was incorporated into the bone, yielding delayed images (Figure 6A–C). Prior to scaffold implantation in the tibia, no delayed 99mTc-HDP image was evident at 0 weeks. However, delayed images were detected at 4 and 8 weeks after implantation. Additionally, both counts in each ROI and the uptake ratio exhibited an upward trend with time. Notably, the uptake ratio decreased between 4 and 8 weeks after implantation (Figure 6D). This suggests rapid osteogenesis during the initial 4 weeks of implantation, followed by bone maturation from 4 to 8 weeks post-implantation. In a study by Zhou et al., the uptake ratio of 99mTc-methylene diphosphonate increased over time in a rabbit model with ulnar defects packed with a porous β-TCP scaffold, although the rate of increase slowed from 8 to 12 weeks post-surgery [19,20]. Conversely, our scintigraphic evaluation indicated a more rapid repair of the beagle tibia’s segmented defect. This finding aligns with our radiographic assessment.

### 2.4. Micro-CT Evaluation

Figure 7 depicts micro-CT images and a dynamic histology of the defect with the scaffold at 8 weeks post-implantation. New bone formation within the structure, primary macro-pores, secondary micro-channels, and tertiary submicron holes, accompanied by complete healing, was observed when the defect site was filled with HA/Al bi-phasic scaffolds. Micro-CT 3D volume images exhibited the regeneration of dense bone within the scaffolds, rendering the scaffold–bone boundary indistinguishable. This observation signifies the successful engraftment of the scaffolds onto the bone. Table 2 displays the quantified bone volume for defects filled with the scaffold and normal bone, as determined by micro-CT analysis (Skyscan software CTAn v.1.18). Measurements of BMD and BV/TV in normal bone and defects filled with the scaffold yielded values of 0.80 ± 0.01, 52.71 ± 0.02, and 0.68 ± 0.04, 51.75 ± 2.21, respectively. These values indicate no significant differences in bone density and BV between defect sites filled with scaffolds and normal bone.

In long-term animal studies involving materials such as HA/β-TCP, coral, and β-TCP for segmental defect restoration, unions have been identified at the host bone–implant interface. However, the mid-region of these materials exhibited fibrous tissue deposition [21,22]. When utilizing only materials without therapeutic agents such as growth factors, native cells from host tissue migration into the scaffolds is essential to generate new tissues. Remarkably, we observed union at the host bone–implant interface and the reunion of the segmented bone defect in this study, accompanied by bone formation through the scaffolds at 8 weeks post-implantation (Figure 7B).

### 2.5. Histological Analysis

Representative sections, after histological processing and staining, for connective tissue and mineralized tissue are shown in Figure 7B and Figure 8 after 8 weeks of implantation. Hollow conical bone growth fronts were observed to regenerate into the defect space from both of the native cortical interfaces adjacent to the defect space, as clearly observed in the empty control defect (Figure 5(A8)). The HA/Al bi-phasic scaffold-implanted experimental group showed a significantly greater bone formation after 8 weeks when compared to the 4-week samples (Figure 5(B4,B8)). After 8 weeks, seamless bone regeneration was evidenced from the host bone (Figure 8A, gray area: left side of the yellow dotted line) into the HA/Al bi-phasic scaffold area (Figure 8A, dark area: right side of the yellow dotted line). A large population of osteocytes, and active interruption among osteocytes, were clearly demonstrated in the newly generated bone region (dark area), while a lesser population and interruption in the host bone region (bright area) were noted (Figure 8A) [23,24]. For the dynamic histology analysis of bone regeneration at 6 and 8 weeks after implantation of the bi-phasic scaffold, alizarin red (Figure 8C) and calcein green (Figure 8D) were injected at designated times. The dark area surrounded by the red line (alizarin red) was proven to be newly formed bone between 0 and 6 weeks after the implantation of the scaffold in the defect area. Another smaller dark area, surrounded by both the red line (alizarin red) and the green line (calcein green), was proven to be newly formed bone between 6 and 8 weeks after the implantation of the scaffold.

Figure 9 represents clear evidence of bone regeneration within the macro-pore spaces and micro-channels of the scaffold. A newly formed bone matrix is observed at the bottom of the image, with embedded osteocytes (Ocy) and a continuing osteoid (OT) at the newly formed bone front. Additionally, osteoblast lining cells (OBs) are well connected on top of the osteoid. Vascularization is the most critical event in the bone tissue development process for healthy and thriving bone regeneration. Numerous red blood cells (RBCs) were observed within the spaces of the macro-pores and even inside the micro-channels, indicating vessel formation [25,26,27]. Remarkably, this single histology slide captures all the events that need to occur during the new bone development process.

Figure 10A depicts a zoomed-in view of a single micro-channel within a strut. Despite the small micro-channel sizes (40–70 μm in diameter), enormous OBs and RBCs migrated into the micro-channel, resulting in new bone matrix filling. Figure 10B illustrates the solid formation of new blood vessels surrounded by a newly formed bone matrix, while ongoing osteogenesis occurs within the macro-pore.

## 3. Discussion

Bone tissue engineering aims to restore function and regenerate tissue by using synthetic materials to fabricate graft substitutes. It is targeted as an alternative to address the increasingly unmet demand for autologous bone grafts. The objectives of this research were to optimize the design of three-dimensional (3-D) porous scaffolds to increase their mechanical strength and to test the feasibility of a synthetic bone graft during in vivo implantation.

Hydroxyapatite (HA) has emerged as a promising material as a bioactive material in the field of bone tissue engineering and regeneration. Specifically, HA porous scaffolds represent a promising strategy for bone regeneration, leveraging their biocompatibility, osteoconductivity, and versatility. However, the low mechanical strength of pure HA porous scaffolds hampers their application in load-bearing regions such as long bones. Alumina (Al) has also emerged as a promising material as a bioinert material in the field of orthopedics and dentistry, with applications such as femoral heads and dental implants. In this study, we propose a bi-phasic porous scaffold composed of Al as the main framing material to lend mechanical strength and HA as a coating material to ensure biological activity for bone regeneration.

HA boasts precisely defined physical and chemo-crystalline attributes, along with a high purity and uniform chemical composition [28,29], setting it apart from other bone substitutes like collagen scaffolds. This inherent characterization, purity, and uniformity enables the reliable prediction of HA’s biological reactivity. Furthermore, the creation of a macro–micro–submicron porous structure with robust interconnectivity facilitates swift and unobstructed blood circulation, facilitating the collection of heterogeneous cells and growth factors from the host bone [30]. The importance of the interconnectivity of scaffolds has been demonstrated in our previous research, where successful bone regeneration, along with vascularization, was observed in highly porous HA scaffolds after 12 weeks’ implantation in a canine mandible defect model [31].

The scaffold’s architecture plays a crucial role in determining its efficacy for bone regeneration. The SEM and micro-CT images reveal a highly porous structure with interconnected pores and micro-channels, mimicking the trabecular bone’s natural architecture. This design facilitates nutrient diffusion, cell migration, and vascularization, essential for promoting new bone formation. Its porous architecture, with interconnected trabecular bone-like structures and multiple pore levels, contributes to a high collective porosity overall [32]. XRD analysis confirmed the presence of both alumina and HA phases, validating successful scaffold fabrication. This thorough characterization underscores the scaffold’s potential for effective bone regeneration.

The successful integration of the scaffold with the host bone is pivotal for achieving functional bone regeneration. The implantation of the scaffold in surgically induced tibial defects resulted in sustained vitality and robust bone regeneration. Radiographic imaging showcased progressive bone healing, with complete defect reunion observed at 8 weeks post-surgery. Moreover, the absence of inflammation or infection underscores the biocompatibility of the scaffold material. In contrast, the control group exhibited minimal bone growth, emphasizing the scaffold’s efficacy in promoting bone formation and defect repair.

Scintigraphy provides valuable insights into bone metabolism and vascularization post-scaffold implantation. The utilization of 99mTc-HDP scintigraphy revealed an efficient bone metabolism and vascularization post-scaffold implantation. The uptake ratio trends suggested rapid osteogenesis followed by bone maturation, aligning with the radiographic assessments [33]. This non-invasive evaluation technique provided valuable insights into the scaffold integration and bone-healing dynamics over time.

Micro-CT analysis further validates the efficacy of the scaffold in promoting bone regeneration. Micro-CT imaging confirmed dense bone regeneration within the scaffold, indicating a successful scaffold–host bone integration [34]. A quantitative analysis demonstrated a comparable bone density and volume between scaffold-filled defects and normal bone, highlighting the scaffold’s ability to support robust bone formation without compromising structural integrity. The seamless integration of the scaffold with the host bone and the absence of fibrous tissue deposition suggest favorable long-term outcomes.

The histological analysis provides detailed insights into the cellular and molecular processes underlying bone regeneration. An histological examination revealed extensive bone formation within the scaffold, with a seamless integration between host and scaffold-derived bone [35,36]. The presence of osteocytes, osteoblasts, and vascular structures within the scaffold confirms active bone formation and remodeling. A dynamic histology using fluorochrome labeling elucidated the timeline of bone regeneration, showcasing progressive bone formation over time [37]. Vascularization within the scaffold macro-pores and micro-channels further underscored the scaffold’s capacity to support physiological bone tissue development.

The promising outcomes of this study hold significant implications for clinical bone repair and regeneration. The HA/Al bi-phasic scaffold exhibits excellent biocompatibility, osteoconductivity, and osteogenic potential, making it a viable candidate for treating segmental bone defects. Further research could focus on optimizing the scaffold’s properties, such as pore size and surface topography, to enhance cellular responses and accelerate bone healing. Additionally, long-term studies are warranted to assess the scaffold’s durability and biomechanical stability in load-bearing applications.

Overall, an integrated approach encompassing scaffold fabrication, surgical implantation, and multi-modal evaluation techniques demonstrates the feasibility and effectiveness of HA/Al bi-phasic scaffolds for promoting bone regeneration. Its precisely engineered porous structure, coupled with favorable biological reactivity and vascularization potential, demonstrates significant advantages over conventional bone substitutes. The successful translation of these findings from preclinical models to clinical settings could potentially revolutionize bone tissue engineering strategies, offering effective solutions for bone defect repair and regeneration. These findings contribute to the ongoing advancements in tissue engineering and hold promise for addressing challenging orthopedic conditions associated with bone loss and trauma.

## 4. Materials and Methods

### 4.1. Fabrication and Characterization of Bi-Phasic Scaffold

The porous HA/Al scaffolds were created through a template-coating method, following a previously established procedure [30]. Initially, a polyurethane sponge sourced from Foam Factory (Macomb, MI, USA), featuring 80 pores per inch, was coated using a slurry comprising nano-alumina powder (AdValue Technology, Tucson, AZ, USA) mixed in distilled water. Enhancing the sintering process and stabilizing the scaffold structures involved introducing binders, including 3% medium-molecular-weight polyvinyl alcohol, 0.5% carboxymethylcellulose, and 1% ammonium polyacrylate dispersant into the slurry mixture. The final slurry ratio of powder/water was 1.5 for coating. Subsequently, the coated sponges were left to dry at room temperature overnight before being sintered at 1500 °C for a duration of 3 h. Thereafter, the HA slurry was prepared in the same manner, but the powder/water ratio was controlled at 0.7 for the HA coating, followed by drying and re-sintering at 1230 °C for 3 h. The overall dimension of the HA/Al scaffold was 1 cm in diameter, 3 mm in the inner cavity, and 2 cm in length (Figure 1). A NeoScopeTM scanning electron microscope (JEOL) was utilized to examine the overall scaffold structure and measure the dimensions of inner micro-channels and submicron holes on the surface, operating at a voltage of 12 kV. The crystalline phase was analyzed using an X-ray diffractometer (XRD, Dmax-2000, Rigaku, Japan) and energy dispersive X-ray spectrometry (EDS, IXRF 5500, Thermo Fisher, Waltham, MA, USA), with a scanning range of 20°–55° in 2θ and a step size of 0.02°. Mechanical testing of the scaffolds was performed in a hydrated state on an Insight 5 test frame (MTS, Eden Prairie, MN, USA) in displacement control mode at a constant strain rate of 0.125 mm/min. Cylindrical scaffolds (16 mm in length and 8 mm in diameter, n = 6) were prepared in order to conform to the 2:1 aspect ratio specified in the ASTM D695 compression testing standard. To prepare the sintered sample for clear cross-section imaging, the sample was embedded in the EpoKwick FC Fast Cure Epoxy System. After curing, it was sectioned using an ISOMET 1000 (Buehler, Germany) and then subjected to surface polishing with 600, 800, and 1200 grit SiC grinding paper, using an Auto Met 250 (Buehler, Germany). The porosity assessment was conducted on non-resin embedded scaffolds using a gas pycnometer (AccuPye II, Micromeritics, Norcross, GA, USA).

### 4.2. Surgical Procedure

In this study, a total of 6 healthy male beagle dogs aged 2 years, with an average weight of 10.33 ± 0.78 kg, were utilized. The Institutional Animal Care and Use Committee of Chungbuk National University approved the experimental protocol (IACUC approval number: CBNUA-138-1001-01). Under general anesthesia, Zoletil 5 mg/kg and Xylazine 2 mg/kg, an aseptic surgical preparation was performed on the left tibia of each dog. The medial side of the tibia was incised to expose the bone, encompassing the skin, muscle, and periosteum. An oscillating saw was employed to create a 2 cm segmental defect through the tibial diaphysis. Among the dogs, 2 were allocated to the control group with unfilled defects, while the remaining 4 had their defects filled with HA/Al bi-phasic scaffolds. Fracture fixation was achieved using a plate and compression screws, followed by suturing. After surgery, Cefazolin 20 mg/kg, Meloxicam 0.1 mg/kg was administered for 7 days. A cast and Elizabethan collar were also employed for 2 weeks. The progress of bone regeneration within the HA/Al scaffolds was monitored at 0, 2, 4, 6, and 8 weeks using X-ray.

### 4.3. Radiographical Assessment

Radiographic assessments were conducted using real-time X-ray images obtained at 0, 2, 4, and 6 weeks post-operation, with animals under general anesthesia, receiving a 3% pentobarbital sodium (30 mg/kg) intravenously, without the need for animal sacrifice. However, samples from the 8-week post-operation period were harvested after the animals were sacrificed to evaluate bone regeneration. The X-rays were generated by an X-ray machine (Toshiba, Tokyo, Japan) positioned at 100 cm. The settings were 60 kVp and 300 mA, with a 0.03 s exposure time. Additionally, computed tomography (CT) scans were obtained at the 8-week mark using a single-slice spiral CT machine (Hi Speed CT/e, GE Medical Co., Chicago, IL, USA). The CT scans were conducted at 120 kVp (130 mA), with a slice thickness of 1 mm and a voxel matrix of 512 × 512. The acquired CT images were processed using 3D imaging software (mimics 13.1, Materialise Co., Leuven, Belgium).

### 4.4. Scintigraphic Evaluation

Canine subjects were administered Zoletil (5 mg/kg, subcutaneously) and Xylazine (2 mg/kg, subcutaneously) for anesthesia. Bone scintigraphy was conducted at 0, 4, and 8 weeks after the surgical procedure. Imaging was conducted using a parallel-hole collimator, utilizing a 20% energy window centered around the 140 keV photopeak of 99mTc, and employing 256 × 256 matrices. The scintigraphy of both tibias was carried out two hours following the intravenous injection of 10 mCi of 99mTc-HDP, using a large-field gamma camera equipped with a high-resolution, face-down, low-energy collimator. A standardized count of 200,000 counts was collected for each tibia image. Regions of interest (ROIs) were delineated within a 1.3 × 0.7 cm box positioned 4–5 cm below the tibial plateau. The total counts within the ROIs were documented for every study conducted on each experiment. Ratios were determined by comparing the percentage of the ROI evaluation with the ROIs for the unaffected contralateral tibia.

### 4.5. Micro-CT Evaluation

Micro-CT was employed to generate reconstructed three-dimensional images of the defects, with or without a bi-phasic scaffold in the defects, over an 8-week period after implantation. The imaging was conducted using a Skyscan Desktop Micro-CT 1172 (Aartselaar, Belgium), operating at a source voltage of 60 kV and a current of 167 μA, with a resolution of 26.7 μm. X-ray radiographs were captured as the specimen rotated 180°, in increments of 0.6°, on a stage. Subsequent to scanning, cross-sectional slices were reconstructed, and each scan was processed using threshold values ranging from 0.008 to 0.031 to differentiate between bone and air. The analysis was carried out utilizing Skyscan software CTAn v.1.18. The evaluation of bone mass and micro-architecture parameters, including bone mineral density (BMD), bone volume (BV), tissue volume (TV), bone surface (BS), trabecular thickness (Tb.Th), trabecular number (Tb.N), and structure model index (SMI) was performed using the built-in software (v.1.18) of the micro-CT.

### 4.6. Histological Evaluation

To assess the dynamic fluorescence histomorphometry of new bone regeneration over time, we administered 25 mg/kg of alizarin red at the 4-week post-surgery mark and 5 mg/kg of calcein green at the 6-week post-surgery mark for a dynamic histology analysis of bone regeneration. After 8 weeks post-implantation surgery, euthanasia was performed, and all the samples were promptly fixed in 10% formalin for 48 h at room temperature. Then, they were subjected to a dehydration and infiltration method for 14 days using a tissue processor (Leica TP1020 System) and a series of graded ethanol solutions (e.g., 70%, 80%, 95%, and 100% for 3 days of each step and 100% for 2 days). The samples were then embedded in photocuring resin (Technovit 7200 VLC, EXAKT, Oklahoma City, OK, USA) for 48 h and polymerized using a light polymerization system (EXAKT 520, EXAKY, Oklahoma City, OK, USA) for 24 h. Block samples were adhered to a parallel plexiglass slide, using the Technovit 7210 VLC system (EXAKT, Oklahoma City, OK, USA). Subsequently, the samples were cut (150–200 µm) using a diamond precision parallel saw (BUEHLER, Isomet 4000, EXAKT, Okalhoma, OK, USA). The cut slides underwent grinding (30–40 µm) and polishing (EXAKT 400, EXAKT, Oklahoma City, OK, USA). An histomorphometric analysis was then employed to quantify the data, using Bioquant. The primary measured parameters included the bone area inside the scaffold, original tibia area, bone area outside the scaffold, and scaffold area. Hematoxylin–eosin and counterstaining with Masson’s trichrome were utilized for the histology images.

### 4.7. Statistical Analysis

All data are reported as means ± standard deviation (SD). The significance in the histological, micro-CT, and mineral density measures reported was determined using a two-way analysis of variance (ANOVA) and Tukey’s test for post hoc evaluation. The significance level was set at *p* < 0.05 for all statistical measures reported.

## 5. Conclusions

This study aimed to assess the viability of a bi-phasic HA/Al ceramic scaffold as a synthetic bone graft in a canine tibia model with a 2 cm segmented bone defect. The scaffold was intentionally designed with a three-tiered structure, comprising macro-pores, micro-channels, and submicron holes, to facilitate cell migration and fluid flow within the scaffold. The creation of this interconnected structure was successfully achieved using a template-coating technique.

Notably, the bi-phasic scaffolds played a crucial role in promoting the integration of the host tissue with the scaffold, spanning both the distal and proximal ends of the defect. Additionally, these scaffolds facilitated the even distribution of newly formed bone throughout the scaffold after an 8-week post-implantation period.

In summary, the study’s findings provide valuable insights into the scaffold’s structural, biological, and functional characteristics, laying the foundation for further research and clinical translation into the fields of regenerative medicine and orthopedic surgery.

## Figures and Tables

**Figure 1 ijms-25-04604-f001:**
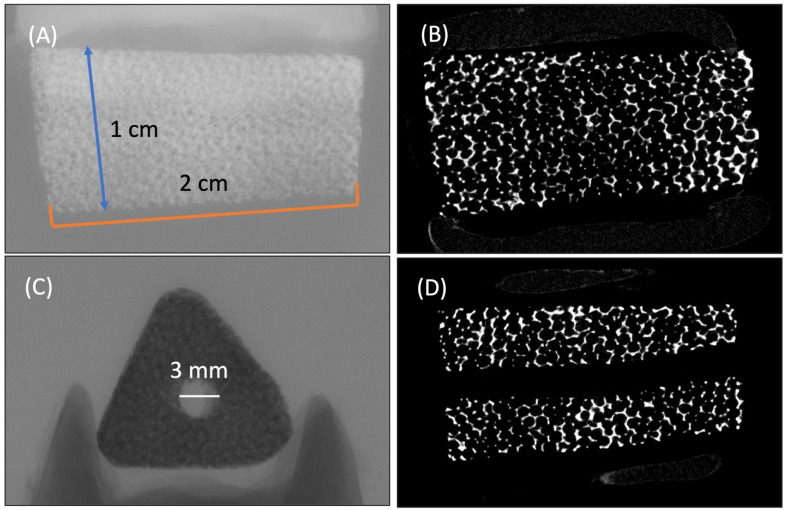
Various micro-CT images of HA/Al bi-phasic scaffold with dimensions. (**A**) Overview of scaffold, (**B**) interconnected macro-pore structure, (**C**,**D**) 3 mm-in-diameter inner cavity structure.

**Figure 2 ijms-25-04604-f002:**
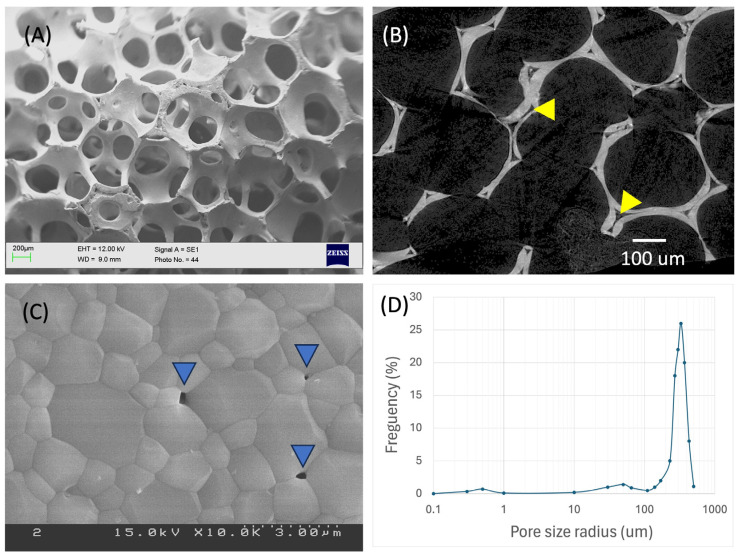
Interconnected, three different structures in the scaffold body. (**A**) Interconnected macro-pores with trabecular-like structure, (**B**) micro-channels that exist inside of each trabecular septum (yellow arrowhead), (**C**) submicron holes for cells to anchor to that exist on the surface of the trabecular septum (blue arrowhead), (**D**) pore size distribution.

**Figure 3 ijms-25-04604-f003:**
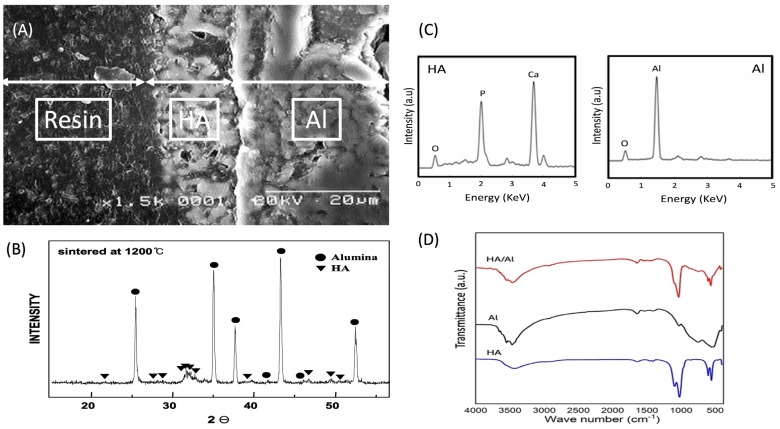
SEM observation after sintering (**A**), X-ray diffraction (XRD) analysis (**B**), EDS spectra (**C**), and FT-IR spectra (**D**) of HA/Al bi-phasic scaffold.

**Figure 4 ijms-25-04604-f004:**
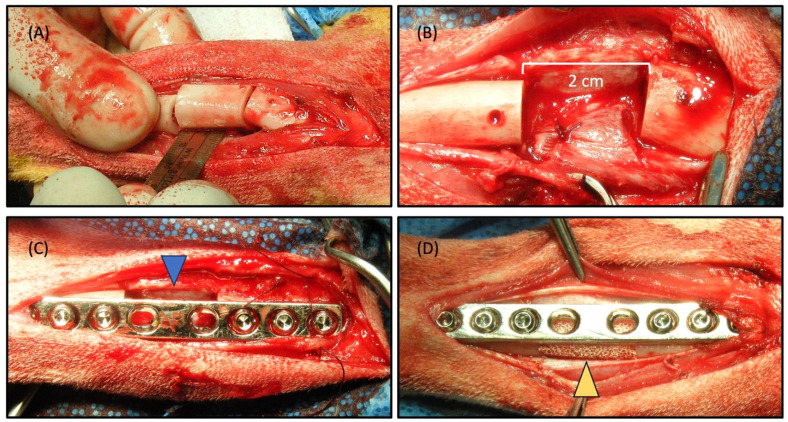
Surgical procedure. (**A**,**B**) creation of a 2 cm segmental defect in the beagle tibia, (**C**) control without scaffold (blue arrow head pointed 2 cm defect only), (**D**) experiment subject with HA/Al bi-phasic scaffold implantation (yellow arrow head pointed 2 cm defect filled with scaffold).

**Figure 5 ijms-25-04604-f005:**
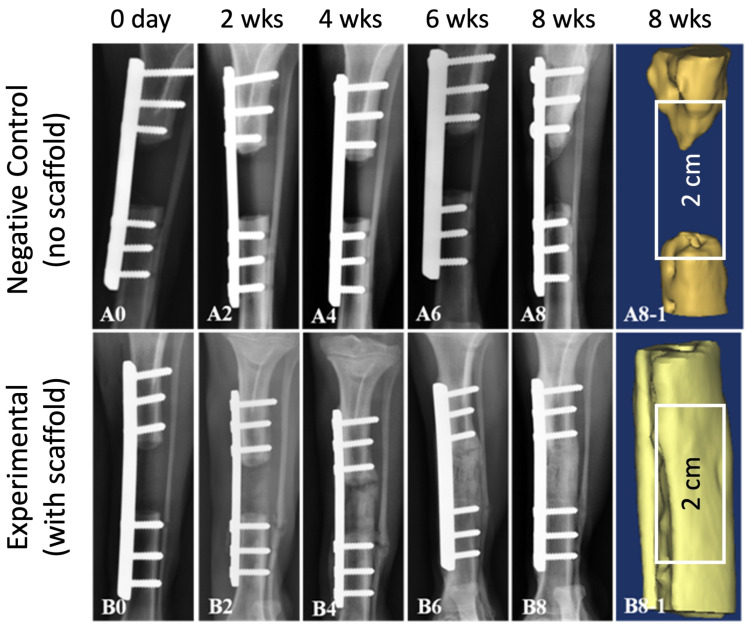
Radiographs of segmental defect site in beagle tibia after 0, 2, 4, 6, and 8 weeks. (**A0**–**A8**) negative control without scaffold, (**A8-1**) reconstructed 3D micro-CT image of the defect site without scaffold at 8 weeks after surgery, (**B0**–**B8**) experiment subject treated with HA/Al bi-phasic scaffold, (**B8-1**) reconstructed 3D micro-CT image of the defect site treated with scaffold at 8 weeks after surgery.

**Figure 6 ijms-25-04604-f006:**
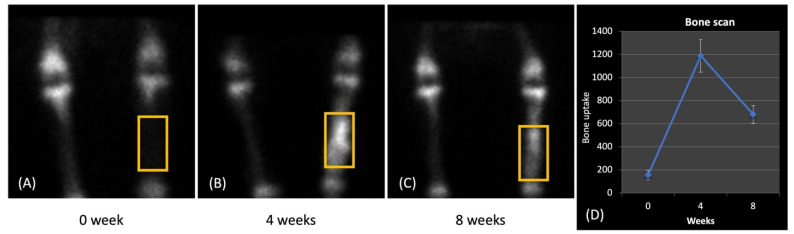
Scintigraphic evaluation at 0 (**A**), 4 (**B**), and 8 weeks (**C**) after surgery for bone uptake (**D**) into the HA/Al bi-phasic scaffold that was implanted in the segmental defect site for treatment.

**Figure 7 ijms-25-04604-f007:**
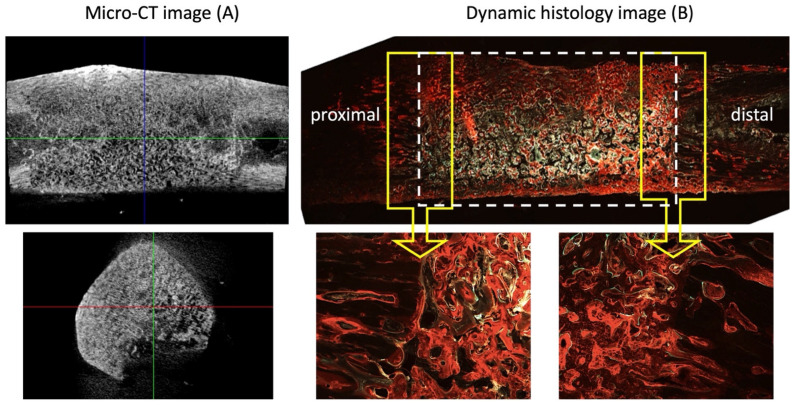
Complete reunion Micro-CT image of 2 cm segmental defect in the beagle tibia by HA/Al bi-phasic scaffold at 8 weeks after surgery (**A**). Segmental defect region is denoted with rectangular white dotted line (**B**).

**Figure 8 ijms-25-04604-f008:**
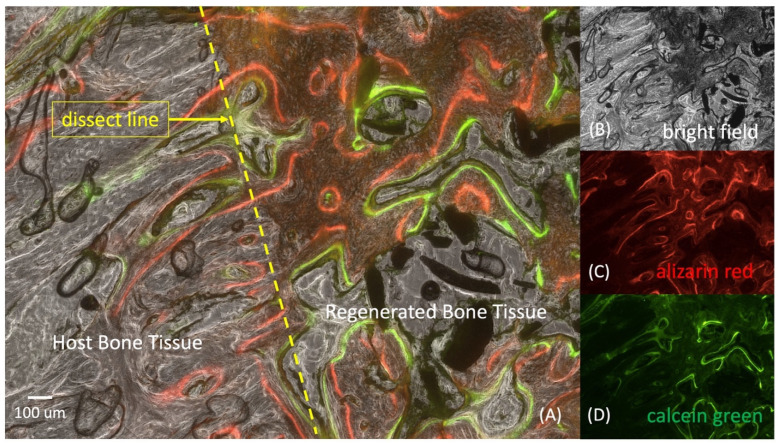
Seamless bone regeneration was evidenced from the host bone (gray area, left side of the yellow dotted line) into the HA/Al bi-phasic scaffold area (dark area, right side of the yellow dotted line) (**A**). Zoomed in bright field image (**B**). Dynamic histology was performed using alizarin red for 6 weeks (**C**) and calcein green for 8 weeks after surgery (**D**).

**Figure 9 ijms-25-04604-f009:**
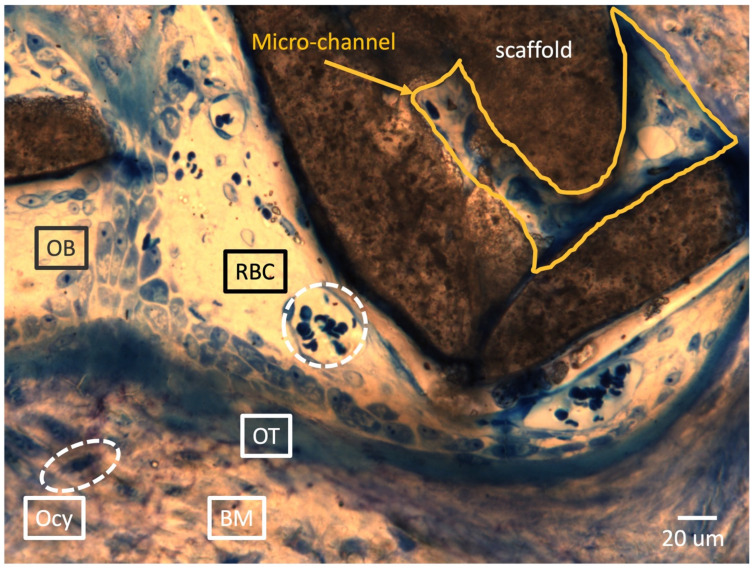
Evidence of bone regeneration into macro-pore space and micro-channel of HA/Al bi-phasic scaffold. OB: osteoblast lining cell, RBC: red blood cell, OT: osteoid, Ocy: osteocyte, and BM: new bone matrix.

**Figure 10 ijms-25-04604-f010:**
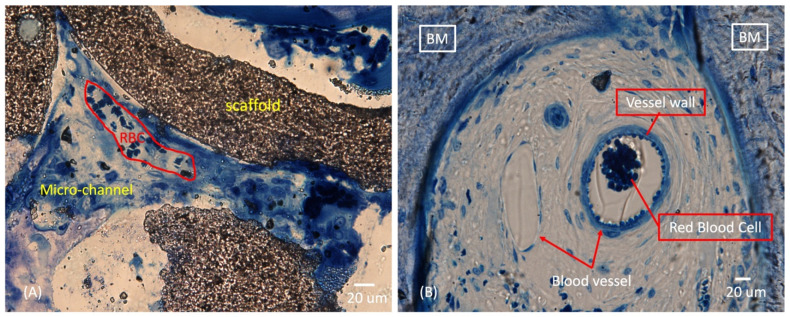
Evidence of vascularization in macro-pore space and micro-channel of HA/Al bi-phasic scaffold (**A**,**B**). RBC: red blood cell and BM: new bone matrix.

**Table 1 ijms-25-04604-t001:** Radiographic bone healing index (mean ± SD).

Group	Weeks after Surgery				
	0	2	4	6	8
Control	0	0	0	0	0.92 ± 0.33
HA/Al scaffold	0	1.33 ± 0.34	3.11 ± 0.41	4.33 ± 0.54	4.91 ± 0.21
	Score	Description			
	0	No visible new bone formation			
	1	Minimal disorganized new bone			
	2	Disorganized new bone bridging graft to host at both ends			
	3	Organized new bone of cortical density bridging at both ends			
	4	Loss of graft–host distinction			
	5	Significant new bone and graft remodeling			

**Table 2 ijms-25-04604-t002:** Bone parameter values of defect sites filled with bi-phasic scaffold at 8 weeks post-implantation (mean ± SD).

Group	BMD (g/cm^3^)	BV/TV (%)	BA/BV (mm^−1^)	Tb.Th (mm)	Tb.N (1/mm)	SMI
Normal Bone	0.80 ± 0.01	52.71 ± 0.02	57.06 ± 0.24	0.67 ± 0.02	1.22 ± 0.11	−7.03 ± 0.37
HA/Al scaffold	0.68 ± 0.04	51.75 ± 2.21	55.43 ± 0.58	0.60 ± 0.07	0.87 ± 0.17	−3.47 ± 0.65

BMD (bone mineral density), BV/TV (percent bone volume), BS/BV (bone specific surface), Tb.Th (trabecular thickness), Tb.N (trabecular number), SMI (structure model index).

## Data Availability

All data used in this article are derived from public databases, and proper citations have been included as required.

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
