# Peer review of "Reconstruction of Segmental Bone Defect in Canine Tibia Model Utilizing Bi-Phasic Scaffold: Pilot Study"

_ijms, 2024, doi:10.3390/ijms25094604_

Round 1

Reviewer 1 Report

Comments and Suggestions for Authors

The authors investigated feasibility of a synthetic bone graft composed of alumina (Al) and hydroxyapatite (HA) without adding cells or growth factors to restore a segmental bony defect. A highly porous bi-phasic ceramic synthetic bone graft composed of hydroxyapatite (HA) and alumina (Al) was developed. This graft was employed to repair critical segmental defects, involving the creation of a 2 cm segmental defect in a canine tibia. The assessment of bone regeneration within the synthetic bone graft post-healing was conducted using scintigraphy, micro-CT, histology, and dynamic histomorphometry.

The results are interesting and quite promising. however it would be better if the authors

1- tested the mechanical properties of the prepared scaffolds.

2-Additionally, pore size distribution using Hg intrusion porosimetry is essential to be conducted.

3- EDS elemental mapping and EDS spectra of the HA/Al scaffold are recommended. furthermore Ca/P ratio should be specified

4- FT-IR spectra of the HA/Al scaffold should be considered 

Minor comments

1- Many images lack scale bar

2- Discussion is short and poor. In-depth discussion should be included

3- Fabrication and characterization of bi-phasic scaffold section should be improved

Author Response

Dear Reviewer

I deeply appreciate your valuable comments, which have resulted in elevating the quality of the research. I have made corresponding changes for all comments from the reviewers. The changes have been addressed and highlighted in the manuscript. The explanations for the modifications and further statements follow:

1- tested the mechanical properties of the prepared scaffolds.

--> added the data and amended accordingly.

2-Additionally, pore size distribution using Hg intrusion porosimetry is essential to be conducted.

--> added in Fig 2 (D).

3- EDS elemental mapping and EDS spectra of the HA/Al scaffold are recommended. furthermore Ca/P ratio should be specified.

--> added in Fig 3 (C) and amended accordingly.

4- FT-IR spectra of the HA/Al scaffold should be considered.

--> added in Fig 3 (D).

Minor comments

1- Many images lack scale bar

--> added and amended accordingly.

2- Discussion is short and poor. In-depth discussion should be included.

--> amended.

3- Fabrication and characterization of bi-phasic scaffold section should be improved.

--> amended.

Reviewer 2 Report

Comments and Suggestions for Authors

In this paper, authors developed an AI/HA composite bi-phase scaffold for segmental bone defect repair and showed complete healing at 8 weeks after surgery and using different novel approaches to prove bone formation and integration. The results are very good, and presentations are very clear. Following are few minor comments that need to be addressed:

Figure 5: If authors can put group information on left side of the figure and time points after surgery on the top of each images panel, it will be much clear than the current layout.

Line 124: (Figure 5C should be Figure 4C).

The last paragraph of discussion section should be deleted since it is the author instructions from the Journal.

Please correct any inconsistencies in the reference format in the text. Some are before “.” Some are after “.”.

In the method section: Histology analysis, can authors provide more details on how long the formalin fixation was? Authors used non-decalcified tissues, can authors provide how long the tissue was dehydrated and infiltrated? Currently, most bone researchers use decalcified bone tissues for histology. Undecalcified tissues have limitations in staining. Authors said they did H&E and trichome staining, which figure is H&E staining (Figure 9, I did not see red cytoplasm and blue nuclear), and which is Trichrome staining (Figure 10?).  Please state this in the figure legends.

Authors said Alizarin red was injected at 4 weeks and calcein was injected at 6 weeks (line 365-367). However, the authors also said Alizarin red was applied at 6 weeks and calcein green at 8 weeks (Line 378-380). Are these applied after dog was sacrificed on the sections? Usually this is injected lively.  Please clarify. 

Author Response

Dear Reviewer

I deeply appreciate your valuable comments, which have resulted in elevating the quality of the research. I have made corresponding changes for all comments from the reviewers. The changes have been addressed and highlighted in the manuscript. The explanations for the modifications and further statements follow:

Figure 5: If authors can put group information on left side of the figure and time points after surgery on the top of each images panel, it will be much clear than the current layout.

--> amended.

Line 124: (Figure 5C should be Figure 4C).

--> amended.

The last paragraph of discussion section should be deleted since it is the author instructions from the Journal.

--> deleted.

Please correct any inconsistencies in the reference format in the text. Some are before “.” Some are after “.”.

--> amended.

In the method section: Histology analysis, can authors provide more details on how long the formalin fixation was? Authors used non-decalcified tissues, can authors provide how long the tissue was dehydrated and infiltrated?

--> added detail conditions for the histology protocols.

Currently, most bone researchers use decalcified bone tissues for histology. Undecalcified tissues have limitations in staining. Authors said they did H&E and trichome staining, which figure is H&E staining (Figure 9, I did not see red cytoplasm and blue nuclear), and which is Trichrome staining (Figure 10?).  Please state this in the figure legends.

--> Absolutely, you are correct. Our focus lies on the osseointegration between the material and the new bone, as well as the new bone formation within micro-channels. Consequently, we opted for undecalcified histology analysis. However, due to the nature of undecalcified preparation, staining poses a challenge. For each sample, we performed both H&E staining and counterstained trichrome staining. The thickness of the slides is approximately 20-25um. Unfortunately, in this regard, clear H&E staining, as seen in decalcified histology analysis, cannot be achieved.

Authors said Alizarin red was injected at 4 weeks and calcein was injected at 6 weeks (line 365-367). However, the authors also said Alizarin red was applied at 6 weeks and calcein green at 8 weeks (Line 378-380). Are these applied after dog was sacrificed on the sections? Usually this is injected lively.  Please clarify. 

--> Yes, these stains were injected lively at 4 weeks and 6 weeks. To clarify, we removed last sentence.